# Comment on "Comparison of ozone measurement methods in biomass burning smoke: an evaluation under field and laboratory conditions" (Long et al. 2021)

Noah Bernays[1], Daniel A. Jaffe[1,2], Irina Petropavlovskikh[2], Peter Effertz[3]

[1]School of STEM, University of Washington, Bothell, WA 98011, U.S.A.
[2]Department of Atmospheric Sciences, University of Washington, Seattle, WA 98195, U.S.A.
[2]NOAA Global Monitoring Laboratory, Boulder, Colorado 80305, U.S.A
[3]CIRES, University of Colorado, Boulder, CO 80309, U.S.A.

*Correspondence to*: Daniel A. Jaffe (djaffe@uw.edu)

## Abstract

Long et al. (2021) conducted a detailed study of possible interferences in measurements of surface $O_3$ by UV spectroscopy, which measures the UV transmission in ambient and $O_3$-scrubbed air. While we appreciate the careful work done in this analysis, there were several omissions and, in one case, the type of scrubber used was mis-identified as manganese dioxide ($MnO_2$), when in fact it was manganese chloride ($MnCl_2$). This misidentification led to the erroneous conclusion that all UV-based $O_3$ instruments employing solid-phase catalytic scrubbers exhibit significant positive artifacts, whereas previous research found this not to be the case when employing $MnO_2$ scrubber types. While the Long study, and our results, confirm the substantial bias in instruments employing an $MnCl_2$ scrubber, a replication of the earlier work with an $MnO_2$ scrubber type and no humidity correction is needed.

## Introduction

Ozone ($O_3$) is a key hazardous atmospheric pollutant. In the U.S., more than 100 million people live in regions that do not meet the National Ambient Air Quality Standards. Wildfires exacerbate $O_3$ pollution (Crutzen et al. 1979; Crutzen and Andreae 1990; Jaffe et al. 2013; 2020; Brey and Fischer 2016; Gong et al. 2017). Given that smoke contains literally hundreds of compounds, it is important to address possible interferences in measurements of $O_3$. Long et al. (2021) conducted a detailed study of possible interferences in UV measurements of $O_3$, the method most commonly used. In the UV method, $O_3$ is measured at 254 nm in a sample airstream and in an airstream where $O_3$ has been removed, usually by a solid-state catalytic scrubber. Long et al. provide an excellent discussion of this method, which we will not repeat here. However, one of the most important aspects in this measurement is the nature of the scrubber that is used to remove $O_3$. For the scrubber, various companies have used manganese dioxide ($MnO_2$), Hopcalite (a mixture of manganese and copper oxides) and manganese chloride ($MnCl_2$). Long et al. compared multiple UV instruments with an NO chemiluminescence instrument, a method which is presumably free from interferences. Long et al. found a significant bias of 16–24 ppb $O_3$ per ppm of CO in one type of UV $O_3$

analyzer (Thermo-Fisher 49i) that was tested without humidity correction, as compared to the NO-chemiluminescence method. The bias was correlated with smoke tracers, such as CO and total hydrocarbons. Other instruments were tested with a humidity correction and found to have a much smaller bias which Long et al. attributed to the humidity correction. According to Long et al., the scrubber types on these instruments were similar, but in fact they were not, as discussed below, and this leads to significant uncertainty in their conclusions.

Long et al. did not cite our earlier study, Gao and Jaffe (2017). In this work, we conducted a comparison between two UV-based $O_3$ analyzers (Dasibi 1008-RS and Ecotech Serinus 10) and an NO-chemiluminescent analyzer in wildfire plumes at the Mt. Bachelor Observatory (MBO) during the 2015 wildfire season. Gao found no significant bias in the UV analyzers relative to the NO-chemiluminescent analyzer in moderate smoke plumes, up to approximately 1 ppm of carbon monoxide (CO). Both of these UV analyzers used an $MnO_2$ scrubber. The precision and bias of instrumentation used in Gao's study along with the quality assurance methods are outlined in the paper and are sufficient to meet Long's study's data quality objectives. A key question is: why were Long et al.'s results different from Gao and Jaffe's results? We address this question below.

**1 Scrubber type misidentified**

Long cites the Thermo-Fisher Scientific Model 49i series instrument's scrubber type as $MnO_2$ (as do others: Kleindienst 1993; Spicer 2010; Turnipseed 2017). However, according to David Sherwin, a Technical Application Specialist III who has been working at Thermo for 18 years, and Nathan Bernardini, a Technical Application Specialist II who has been working at Thermo for close to 5 years, the scrubbers in the 49, 49c, and 49i series have always used $MnCl_2$, not $MnO_2$. While we have not done chemical tests on the scrubber, we feel that the manufacturer is in the best place to know what is inside their instrument. The names and email addresses of the Thermo-Fisher scientists with whom we communicated, as well as screenshots of our email correspondence, can be found in the author's final comment in the discussion section. Please note that there is no info about the scrubber type in the manual. Due to this scrubber type misidentification, Long et al. did not test any $O_3$ analyzer with a true $MnO_2$ scrubber and without humidity correction, the most common way these instruments are deployed.

**2 Recent data from the Mt. Bachelor Observatory confirm bias with $MnCl_2$ scrubber type**

The Mt. Bachelor Observatory is a high elevation research station in the Pacific Northwest that has been used for many years to study $O_3$ and other pollutants (e.g. Jaffe et al. 2018). Starting in 2018, we have deployed two $O_3$ instruments at the MBO, the Ecotech Serinus 10, previously used in the Gao and Jaffe study, and a Thermo-Fisher 49c, a similar instrument to the one used in Long's study which uses the same scrubber and no humidity correction. Generally, the Ecotech and Thermo-Fisher instruments agree well, but in a particularly strong period of wildfire smoke, we saw a substantial difference in the two measurements. Figure 1 shows data from a 3-week period in September–October 2020, when we experienced heavy smoke at the MBO. The slope (.0112 ppb of $O_3$ per ppb CO)

is smaller but of the same order of magnitude as that reported by Long et al. for comparisons of the Thermo-Fisher
to the NO-chemiluminescent instrument (.016–.024 ppb $O_3$ per ppb CO).

In the absence of smoke, we see good agreement between the two measurements. Figure 2 shows the agreement between the Thermo and Ecotech instruments during non-smoke periods (defined as CO < 200 ppb), with a root mean squared difference of less than 1 ppb. Given our earlier comparison establishing that the Ecotech instrument did not show significant bias (Gao and Jaffe 2017), we contend that these findings corroborate Long's conclusion
that the Thermo-Fisher instrument exhibits a significant positive bias at high CO levels. We believe the $MnCl_2$ scrubber in the 49i is the primary cause for the discrepancy between Long and Gao's findings.

### 3 Nafion dryer vs. scrubber impacts on $O_3$ measurements: need for further research

When Long et al. put a Nafion dryer on their Thermo-Fisher instrument midway through the study, the bias was reduced by an order of magnitude. We agree with Long et al. that the Nafion dryer reduced not only water vapor, but
probably also scrubbed many of the VOC's that were causing the bias. While Nafion is known to transfer $O_3$ and lower molecular weight alkanes efficiently, it will remove more complex VOC's that are likely responsible for the bias in UV instruments (Perma-Pure 2022). Similar tests with/without a Nafion drier were not done for the other instruments. The Nafion-dried 2B-205 instrument in Long's study showed $O_3$ artifacts an order of magnitude lower than the non-dried UV analyzers, but this can be explained by the 2B's $MnO_2$-containing Hopcalite scrubber acting
similarly to a pure $MnO_2$ scrubber. We note that current EPA recommendations are to include Nafion dryers for UV $O_3$ instruments (Halliday et al. 2020), and we see no downside to this recommendation. But given that this remains a recommendation, and to interpret past data, we suggest that future experiments on $O_3$ bias include instruments with a true $MnO_2$ scrubber with, and without, humidity correction, as the most common field setup does not include a drying system.

**Data availability**

Data from the Mt. Bachelor Observatory are archived at the University of Washington's Research Works Archive (https://digital.lib.washington.edu/researchworks/discover?scope=%2F&query=%22mt.+bachelor+observatory%22 &submit=&filtertype_0=title&filter_relational_operator_0=contains&filter_0=data).

**Author contributions**

NB completed most of the data analysis, instrument calibrations and wrote the first draft of the manuscript. DAJ is the Principal Investigator for the Mt. Bachelor Observatory (MBO) and contributed to the overall analysis and manuscript preparation. IP and PE are responsible for the duplicate (Thermo) $O_3$ measurements at MBO and contributed to data interpretation and final preparation of the manuscript.

**Competing interests**

The contact author has declared that neither they nor their co-authors have any competing interests.

**Financial support**

The Mt. Bachelor Observatory is supported by the National Science Foundation (grant #AGS-1447832) and the National Oceanic and Atmospheric Administration (contract #RA-133R-16-SE-0758).

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

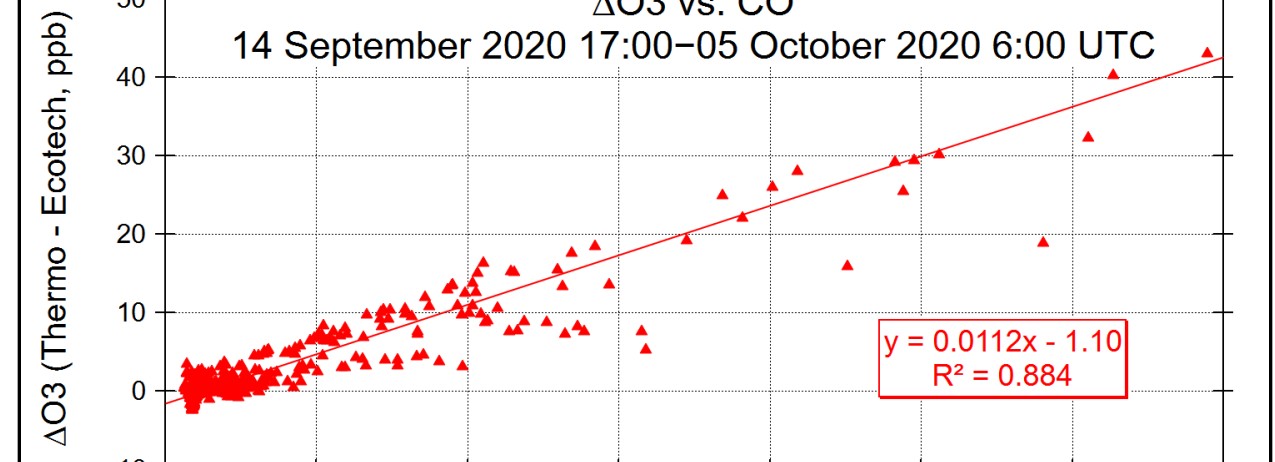


**Figure 1: Difference in O₃ readings between the Thermo-Fisher and Ecotech UV instruments vs. CO for a 3-week period starting 14 September 2020. During this period, the Thermo-Fisher instrument gave readings that were up to 45 ppb higher than the Ecotech instrument. Values are hourly averages.**

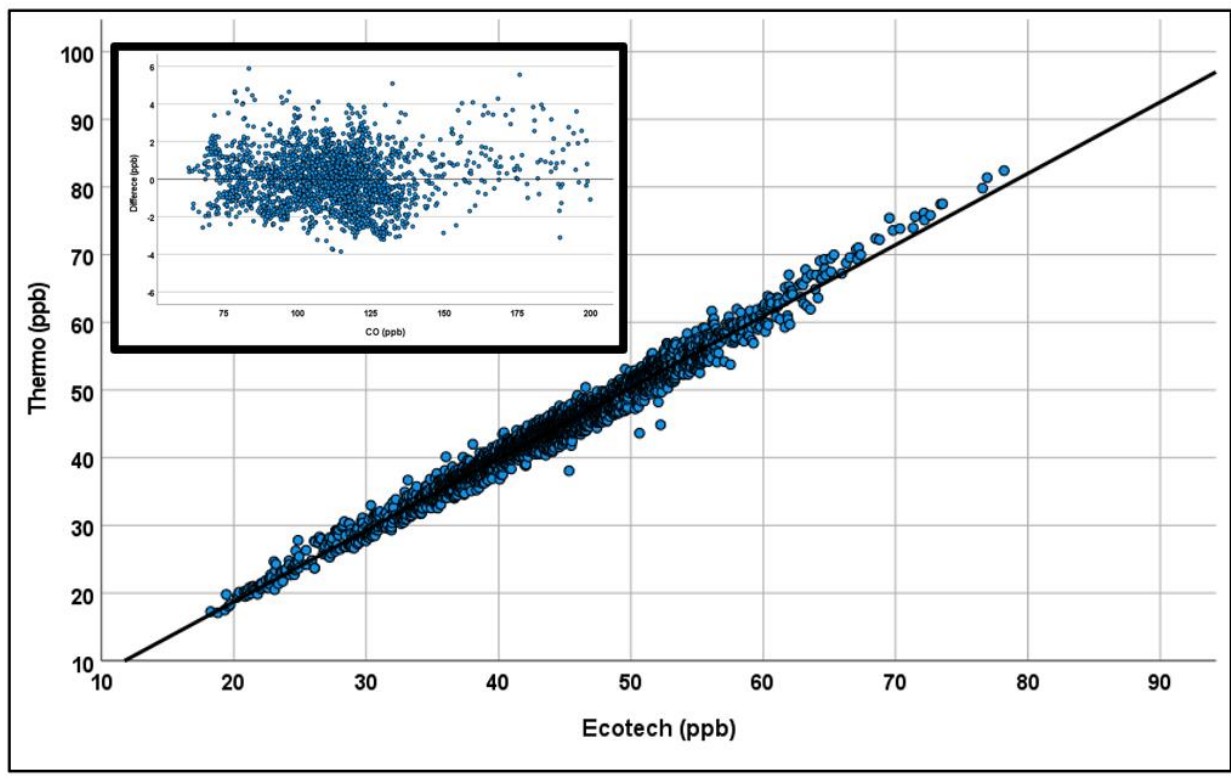


**Figure 2: O₃ measured by the Thermo-Fisher 49c instrument vs. Ecotech instrument at MBO during non-smoke periods (defined as CO < 200 ppb). Data are hourly averages of all valid data for both instruments in 2020. The inset shows a difference plot (Thermo-Ecotech) vs. CO for the same data. The root mean squared error (difference) of the Thermo vs. Ecotech plot is 0.9 ppb, and the linear regression line has a slope of 1.055, a y-intercept of -2.4 ppb, and an $R^2$ of 0.98.**