# Peer review of "Comment on "Comparison of ozone measurement methods in biomass burning smoke: an evaluation under field and laboratory conditions" (Long et al. 2021)"

_Atmospheric Measurement Techniques, 2022_

## Referee Comment (RC1)

**Review** Comment on "Comparison of ozone measurement methods in biomass burning smoke: an evaluation under field and laboratory conditions" (Long et al. 2021)
By Bernays et al.

Biomass burning is becoming an increasingly important component of air quality as fossil fuel combustion becomes cleaner.  It is important to be able to measure the ozone produced by wildfires and agricultural fires.  As such the Long et al. paper calls into question the accuracy of ozone measurements with commercial instruments.  The Comment from Bernays et al. states "the type of scrubber used was mis- identified as manganese dioxide ($MnO_2$), when in fact it was manganese chloride ($MnCl_2$)."  If correct, it is essential to publish this Comment.  My major point in this review is that Bernays et al must offer proof that the Thermo-Fisher 49i uses $MnCl_2$. Last I checked, the composition of the ozone killer was a trade secret.  **A letter from Thermo-Fischer certifying that the composition of their ozone scrubber is or was $MnCl_2$ would make this comment convincing.**

Other points:
Gao et al. (2017) is really Gao and Jaffe (2017) – right?

It is not clear to me that Long et al. correctly accounted for the small humidity sensitivity of NO chemiluminescence for O3 detection. (Boylan et al., 2014).  Bernays et al. should check this out and comment as appropriate.

Reading Long et al. or Bernays et al. one would think that ozone from biomass burning began to be understood in 2004 or even 2013.  Some influential papers are given at the end of this review and should be cited.

[*Boylan et al.*, 2014; *Crutzen and Andreae*, 1990; *Crutzen et al.*, 1979]

Boylan, P., D. Helmig, and J. H. Park (2014), Characterization and mitigation of water vapor effects in the measurement of ozone by chemiluminescence with nitric oxide, *Atmos. Meas. Tech.*, *7*(5), 1231-1244.
Crutzen, P. J., and M. O. Andreae (1990), Biomass burning in the tropics: Impact on atmospheric chemistry and biogeochemical cycles, *Science*, *250*, 1669-1678.
Crutzen, P. J., et al. (1979), Biomass burning as a source of atmospheric gases CO, H2, N2O, NO, CH3Cl, and COS, *Nature*, *282*, 253-256.

---

## Author Comment (AC1)

**We appreciate both reviewers helpful comments.   See below for specific responses.**

**Author's reply to RC1:**

"The Comment from Bernays et al. states 'the type of scrubber used was mis- identified as manganese dioxide (MnO2), when in fact it was manganese chloride (MnCl2).' If correct, it is essential to publish this Comment. My major point in this review is that Bernays et al must offer proof that the Thermo-Fisher 49i uses MnCl2. Last I checked, the composition of the ozone killer was a trade secret. A letter from ThermoFischer certifying that the composition of their ozone scrubber is or was MnCl2 would make this comment convincing."

> Please see the screenshots below of exchanges with two separate Thermo representatives stating that the scrubber is made of MnCl2.  David Sherwin is a Technical Application Specialist III and has been working at Thermo for 18 years. The names and email addresses of the Thermo-Fischer scientists that we communicated with will be incorporated into the final comment.  Please note that there is no info about the scrubber type in the manual.

"Other points: Gao et al. (2017) is really Gao and Jaffe (2017) – right?"

> Yes, Gao et al. (2017) is really Gao and Jaffe (2017).

"It is not clear to me that Long et al. correctly accounted for the small humidity sensitivity of NO chemiluminescence for O3 detection. (Boylan et al., 2014). Bernays et al. should check this out and comment as appropriate."

> It is unfortunate that the authors of the original manuscript (Long et al) chose not to respond. However, we note that in section 2.1.1 of Long et al 2021, they state, "Although there is a known water vapor interference with chemiluminescence technology (Kleindienst et al., 1993), the TAPI T265 uses a Nafion® tube dryer system to remove water vapor from the air prior to making the measurement, thus eliminating any humidity-related effects." Even if their NO instrument did not have a Nafion dryer which eliminated humidity effects, we do not think that the magnitude of any humidity sensitivity would explain the discrepancy they reported between the scrubber-less instruments and those with a solid state scrubber. For that reason, we did not focus on humidity in our response.

"Reading Long et al. or Bernays et al. one would think that ozone from biomass burning began to be understood in 2004 or even 2013. Some influential papers are given at the end of this review and should be cited."

> The reviewer is correct that the O3 from biomass burning was recognized much earlier than we made it seem. We intend to add these citations to the final manuscript."

**Author's reply to RC2:**

"Table 1 in Long et al. lists the Thermo Scientific model 49i monitor that was used in their study as having been equipped with a $MnO_2$ catalyst. This is also mentioned in their text, section 2.1.2. first sentence. To the best of my knowledge, this has been and still currently is the default, and only configuration, in which this monitor can be purchased from Thermo Scientific. I cannot trace from where Bernays et al. got the information that the monitor used by Long et al. had a different, i.e. a $MnCl_2$ scrubber. I don't remember ever seeing an ozone UV absorption monitor that used an $MnCl_2$ scrubber. This is an important piece of missing information. This certainly needs to be clarified by Bernays et al. before their comment should be accepted for final publication."

We agree that past studies have suggested that the $O_3$ scrubber in the Thermo 49 series analyzers is $MnO_2$. However, this is not in the manual and multiple contacts with Thermo Fischer employees confirmed that the scrubber is in fact $MnCl_2$. We have documented our correspondence with them below and will include the names of the Thermo employees in the manuscript. While we have not done chemical tests on the scrubber, we feel that the manufacturer is in the best place to know what is inside their instrument. In any case, we show that the Thermo instrument behaves very differently from another instrument with an $MnO_2$ scrubber. For this reason, we feel it is important to publish our comment.

"Furthermore, Long et al. should be given an opportunity to comment on this question."

Long et al were given the opportunity to respond. It is unfortunate that they chose not to.

**Emails received from Thermo-Fischer concerning the 49c scrubber.**

**Email #1:**

From: David.Sherwin@ThermoFischer.com

To: Noah Bernays (nbernays@uw.edu)

November 12, 2021

***David Sherwin, an 18-year employee of Thermo-Fischer, confirms that the Model 49c and 49i analyzers use a manganese chloride scrubber.***

[Figure]

**Email #2:**

From: Nathan.Bernardini@ThermoFischer.com

To: Noah Bernays (nbernays@uw.edu)

March 3, 2022

*Further verification from a Thermo company representative that past versions of the 49 series analyzers also used a manganese chloride scrubber.*

[Figure]

**New and updated figures we intend to use in the final comment/manuscript:**

The following plot was updated using the newest data from the Mt. Bachelor 2020 yearly dataset:

[Figure]

We intend to add the following new figure to the manuscript which demonstrates good agreement between the Thermo and Ecotech instruments during non-smoke periods.

[Figure]

**O₃ measured by the Thermo-Fischer 49c instrument vs Ecotech instrument at MBO when CO < 200 ppb (non-smoke periods). This includes all valid for both instruments in 2020. The inset shows a difference plot (Thermo-Ecotech) vs CO for the same data. The root mean squared error (difference) is 0.9 ppb and the linear regression line has a slope of 1.055, intercept of -2.4 ppb and an R$^2$ of 0.98**